# Preliminary Studies on Rare Elements Addition and Effect on Oxidation Behaviour of Pack Cementation Coatings Deposited on Variety of Steels at High Temperature

**DOI:** 10.3390/ma14226801

**Published:** 2021-11-11

**Authors:** Tomasz Dudziak, Ewa Rząd, Tomasz Polczyk, Katrin Jahns, Wojciech Polkowski, Adelajda Polkowska, Michal Wójcicki

**Affiliations:** 1Centre for Corrosion Studies, Lukasiewicz Research Network—Krakow Institute of Technology, 73 Zakopianska Str., 30-418 Krakow, Poland; ewa.rzad@kit.lukasiewicz.gov.pl (E.R.); tomasz.polczyk@kit.lukasieiwcz.gov.pl (T.P.); wojciech.polkowski@kit.lukasiewicz.gov.pl (W.P.); adelajda.polkowska@kit.lukasiewicz.gov.pl (A.P.); michal.wojcicki@kit.lukasiewicz.gov.pl (M.W.); 2Steel Institute IEHK, RWTH Aachen University, Intzestraße 1, 52072 Aachen, Germany; Katrin.Jahns@iehk.rwth-aachen.de

**Keywords:** pack cementation, oxidation, rare elements, XRD, SEM, EDS

## Abstract

The aim of the paper was to investigate the air oxidation behaviour of pack aluminised steels exposed at 650 °C for 1000 h in static natural air atmosphere. The pack coatings were doped by rare elements such as gadolinium (Gd), cerium oxide (CeO_2_), and lanthanum (La) in order to enhance the corrosion resistance and plasticity of the deposited layers. In this work, the following steels were used: 16M, T91, VM12, Super 304H, and finally SANICRO25. The results indicated a much higher corrosion resistance in the coated 16M, T91, and VM12 steels; the steels with a higher Cr content than 16 wt % Cr indicated a better behaviour in the uncoated state than in the coated state. However, the observed difference in mass gain between the uncoated and the coated austenitic steels was not enormous. Furthermore, the addition of RE elements to the coating showed some effect in terms of coating thicknesses and differences in the layer structures. The materials prior to testing and after the exposure were investigated using XRD, the SEM X-ray maps with an EDS instrument were used for particular samples to evaluate the phase identifications, element concentrations, microstructure, and chemical composition.

## 1. Introduction

There are many different steel grades used in the power plant industry to secure energy production. Depending on the Cr content in the metal matrix of a steel, steels are used in different temperature regimes to withstand a long service duration. In the coal power industry, the exposure temperature may reach 650–750 °C in the hottest sections such as super heater (SH) or re-heaters (RH). Here, steels with remarkable mechanical properties and high-temperature oxidation resistance must be installed, while in the other sections where much lower temperatures are met (up to 500–550 °C), the steels with up to 10 wt % Cr such as 16Mo3, T22, T91, and VM12 may be found. There are several steels with adequate oxidation resistance such as 304H or 316L to be used in the hottest sections of a power plant. However, apart from a high degree of oxidation resistance and mechanical properties (creep resistance), these steels also show a reduced thermal conductivity in contrast to low-alloyed steels with reduced oxidation resistance and mechanical properties (creep). In order to overcome the problem of the reduced oxidation resistance of low alloyed steels at high temperatures, diffusion coatings produced by pack cementation are an idea to develop new systems for the high-temperature protection of boiler steels. The pack cementation process was invented years ago to protect gas turbines with Al-rich coatings [1]; however, further studies indicated that diffusion coatings are susceptible to cracking due to the formation of brittle intermetallic phase formation such as Fe_2_Al_5_. Further development toward the reduction of brittle phase formation concentrated around the transformation of Fe_2_Al_5_ into FeAl phase by additional heat treatment at high temperature in Ar atmosphere, as reported by Xiang and Datta [2]. Nevertheless, due to a high degree of brittleness, crack formation during operational services, and the development of other technologies such as thermal spray [3], pack cementation coatings are currently not under extensive research. However, the appearance of rare elements (RE) in material science such as cerium (Ce), yttrium (Y), lanthanum (La) and gadolinium (Gd) or their oxides opened new possibilities for pack cementation coatings. The amount of publications regarding the effect of RE on the coatings’ microstructure, thermal stability, corrosion resistance, and mechanical behaviour is numerous; Y added to a laser cladding NiAl-based coating in a range of 1.5–3.5 at % reduces the hardness and anti-attrition of the cladding layer and improves obviously its wear and oxidation resistances [4]. Furthermore, to increase the inlet temperature in an aero jet or gas turbine, Thermal Barrier Coatings (TBC) are used, consisting of an yttria partially stabilized zirconia (ZrO_2_–Y_2_O_3_) top coat and a NiCoCrAlY/PtAl-based metallic bond coat [5]. The role of Y_2_O_3_ in the coating is to stabilize the ZrO_2_ phase against transformation from monoclinic to tetragonal phase at around 1173 °C [6,7]. The addition of CeO_2_ to protective coatings is often attributed to the increased resistance to crack formation under thermal loads, as the formation of cracks is one of the major degradation mechanisms that cost a coating life [8,9]. Reviewing the RE effect on protective coatings at high temperatures, as mentioned earlier, the addition of other RE is particularly concentrated in TBC, with 7–8 wt % yttria-stabilized zirconia (8YSZ). The 8YSZ possesses a metastable tetragonal phase (t’), and Y_2_O_3_ is used to stabilize the ZrO_2_ structure [10].

In case of bare steels or Ni-based alloys, the addition of RE improves the oxidation resistance, especially in cyclic conditions due to the formation of a slow-growing oxide scale [11,12]. Generally, three main mechanisms are proposed to explain the RE effect on cyclic oxidation: (i) the dynamic-segregation theory [13], (ii) reactive elements attracting sulphur [14], and finally (iii) reactive element additions promoting a preferential cationic or anionic diffusion [15,16]. Ishii et al. [17] found that the introduction of reactive elements to FeCrAl alloys decreased the growth rate of the Al_2_O_3_ scale. Pack cementation coatings are still an interesting research topic. In the past, Kipkemoi et al. [18] investigated the effect of HfO_2_ (as a Hf source) on T22 steel cyclic resistance. The study confirmed improvement; however, the diffusion coating was applied at 1050 °C, suggesting degradation of the microstructure and decreasing mechanical properties of low-alloyed steel (T22). Nevertheless, despite the high number of papers related to the RE effect in Fe, Ni-based alloys and coatings published in the past, there is still lack or there is only limited information regarding the effect of RE addition to pack cementation coatings with such additions as Gd, La, and CeO_2_. The significance of the current research is that it is the first to show the RE role in pack coatings on a variety of steels for the energy sector. The presented work will increase knowledge regarding the effect of RE at high temperature. Thus, there is a large gap in the research related to the effect of RE in pack-processed coatings. In this work, the effect of Gd, La, and CeO_2_ addition to pack cementation coatings on oxidation resistance was investigated using several steels: 16M, T91, VM12, SANICRO25, and Super 304H boiler steels.

## 2. Experimental Procedure

### 2.1. Materials

In this work, 5 different steels were used. The materials were selected to cover the steels with a low Cr content less than 1 wt % Cr up to Cr content as high as 25 wt % Cr in a metal matrix. The chemical composition of the steels is shown in Table 1. The chemical composition provided in Table 1 is delivered from certificates; additional EDS analyses were carried out to confirm the chemical composition (not shown here).

The materials prior to coating development were cut into the samples with dimensions of 10 × 10 × 3 mm^3^. The samples prior to coating deposition were grounded with 600 and 1200 SiC paper and cleaned in acetone using an ultrasonic bath at a temperature of 40 °C for 15 min to degrease and clean the surface.

### 2.2. Pack Cementation Process

The pack cementation process is schematically presented in Figure 1 [19]. Pack cementation is in fact an in-situ CVD method whereby the process is established by halide salts (activator) such as NH_4_Cl, NaCl, AlCl_3_, or NH_4_F [20]. The substrate to be coated is inserted into a ceramic crucible with a lid and sealed by high-temperature glue. The sealed material is covered by the inert filler (Al_2_O_3_ or SiO_2_) and specific powder: Al, Cr, Si, or different, depending on the type of the coating that needs to be produced. Generally, the described methodology is called an “in-pack cementation process”, because the coated material is inside the mixture of powders. Alternatively, pack cementation can be modified: when the material to be coated is suspended above the pack mixture, the pack cementation is called the over-pack cementation method or out-of-pack cementation [19].

In this work, the in-pack cementation method was used to develop the protective diffusion coatings. The pack powders were prepared by accurately weighing and mixing the required amounts of powders using a mortar and pestle. The average powder size had a value of 50 µm. The same mixture of powders contained Al (8%), Al_2_O_3_ (88%), and a halide salt AlCl_3_ (4%) was used for all the steels used in this work. The mixture of powders together with a sample prepared for the deposition of a coating was placed in a cylindrical Al_2_O_3_ crucible (CERAMIT, Tłuczań, Poland) with a lid and loaded into an alumina tube inside a tubular furnace(Carbolite, Sheffield, UK) Prior to the deposition of the coating, a tubular furnace was closed by 4 stainless steel screws at room temperature and rinsed with argon for up to 2 h, which flowed (100 mL/min) throughout the tubular furnace to remove gaseous impurities and humid air at room temperature. To secure an air and moist-free atmosphere, an argon flow was established (100 mL/min) at 150 °C for an additional 2 h. The deposition temperature in this work was 750 °C. However, the pack process usually is conducted at much higher temperatures than 750 °C, as was carried out in this study i.e., for Ni-based alloys [21]. Much higher temperatures provoke fast diffusion of the elements from the pack mixture and from the substrate. A long (6–30 h exposure)-lasting deposition process leads to degradation of the microstructure, resulting in serious obstacles in the mechanical properties associated with grain coarsening and the high concentration of carbide precipitation [22,23]. To avoid this, the materials in this work were exposed only at 750 °C for 12 h. During the process, a constant argon flow of 100 mL/min was maintained; the argon flow was kept during the cooling down period to room temperature to avoid oxidation processes. Every sample produced in this project was weighed prior to the deposition process and afterwards using a high-accuracy electronic balance (10^−6^ g) (Sartorius, Göttingen, Germany) in order to estimate the mass gain of the samples and show the coating development on the alloy surface. The addition of individual La, Gd, and CeO_2_ powders was carried out during the pack mixture preparation, whereby the amount of individual rare element did not exceed 1% in the pack mixture. Finally, the coatings were not further heat treated after the deposition process. Thus, the as-received state was used for oxidation tests at 650 °C for 1000 h. The samples exposed in air atmosphere were analysed using XRD from the oxidised surface. Cross-sectional analyses were conducted on selected samples only due to the high number of samples (25 samples). The samples were mounted in a conductive resin perpendicular to the surface using a clip. The sample was firstly grounded using 320, 600, and 1200 SiC paper, and it was further polished using 9, 3, and 1 µm suspension paste based on SiO_2_ to obtain a mirror-like surface. Finally, the samples were dried using acetone to avoid any trails on the freshly polished surfaces. Due to a high number of samples (25 samples in total), codes for the samples were introduced for a better traceability. The codes are shown in Table 2.

### 2.3. Air Oxidation

Figure 2 shows a schematic representation of the rig used for the air oxidation test for 1000 h at 650 °C.

The rig presented in Figure 2 was used to accommodate 25 samples fabricated via the pack cementation process in this study. The samples were exposed for 1000 h at 650 °C. Every 100 h, the samples were removed from the furnace and weighed to achieve the kinetic data. Together with the coated samples with the addition of rare elements, the coated samples with no rare elements were exposed, and the uncoated samples as references were in the rig as well. The ramp rate was set up for 5 °C/min, the samples were placed on an Al_2_O_3_ ceramic holder with shelves to accommodate the high number of samples. The holder with the samples was inserted into the ceramic linear with both sides being closed by ceramic plugs. The plugs were inserted to avoid the access of humid air from an ambient atmosphere.

## 3. Results and Discussion

### 3.1. As-Deposited Diffusion Coatings

The presented data in this paragraph do not show the results of the coatings with RE addition. The respective addition of RE (Gd, La, CeO_2_) up to 1% did not have any effect on the microstructure (phase constituents) or the chemical composition of the as-deposited coatings. Figure 3A shows the as-deposited diffusion coating on 16M steel using the parameters described above (750 °C, 12 h). The coating shows a thickness of 60 µm with a good adherence to the metallic substrate 16M. There is a sharp interface between the substrate and the coating. Perpendicular to the coating, a crack has formed, which was probably due to metallographic sample preparation. The EDS analysis performed on the coated 16M steel showed mostly a single-phase coating with the outermost thin layer with a thickness of 1–2 µm (as indicated in Figure 3A in white rectangular), and the major layer thickness reached around 60 µm. Based on EDS analysis performed on the as-deposited 16M steel, it can be concluded that the outermost layer consists of the FeAl_2_ phase, whereas the major layer was occupied by the Fe_2_Al_5_ phase. According to Rohr et al. [24], these phases are frequently present in low-alloyed steels such as 16M or T22 steel. The steel with 9 wt % Cr (T91) showed the formation of a thinner diffusion coating, as presented in Figure 3B. The T91 steel was coated under the same conditions as 16M steel. However, the coating thickness was much lower (40 μm) than observed in the case of 16M steel. The chemical composition of the coating developed on T91 steel was different to that observed for the 16M steel. The thicker part (30 μm) of the coating consisted of 45 wt % Fe and 55 wt % Al, suggesting the formation of the FeAl_2_ phase according to the Fe-Al diagram [24]. The thinner part of the coating (10 μm) consisted of 60 wt % Fe and 40 wt % Al, indicating an Fe_3_Al phase development (enlarged image in the lower left corner in Figure 3B). The formation of the Fe_3_Al phase in diffusion coatings is quite unique, since mostly, other researchers found the development of FeAl, FeAl_3_, and Fe_2_Al_5_ phases [25,26,27]. Finally, no Cr was found within the coating after diffusion coating deposition. The steel with 12 wt % Cr (VM12 steel) developed a thick and well-adherent coating with a thickness of 100 μm, as shown in SEM image in BSE mode (Figure 3C). A sharp interface between the coating and the substrate was observed together with precipitation of the second phase enriched in Cr (15 wt % Cr) localised within the outermost part of the coating. The major chemical composition of the coating consisted around 40 wt % of Fe and around 40 wt % of Al with some low concentration of Cr (2 wt %), suggesting the formation of the FeAl_2_ intermetallic phase. The second phase developed within the coating consisted of up to 8 wt % Cr. Chemically, the composition of the coating was constant, leading to the conclusion that the FeAl_2_ phase was predominantly developed on VM12 steel based on EDS analysis.

A similar situation to that observed for VM12 steel was found for SANICRO25 steel (Figure 3D), where a thick (75 μm) coating was formulated during the pack process conducted at 750 °C for 12 h. The coating was slightly thinner than that formed in Super 304, but it was much thicker than that observed on 16M and T91 steels. The developed coating showed no cracks, but a 2 µm thick 20 wt % Cr layer was observed between the coating and the substrate. This layer consists as well of 10 wt % Ni. The main chemical constituents within the outermost part of the coating according to EDS analysis performed on the as-received coating was Al (70 wt %), Fe (16 wt %), Cr (8 wt %), and finally Ni (5 wt %). The middle part of the coating showed a similar chemical composition suggesting a constant element flux during coating formation. The bottom part of the coating (nearby 2 µm thick)-enriched Cr layer was depleted in Al (52 wt %) and enriched in Fe (26 wt %) and Cr (16 wt %), whereas the Ni concentration was similar (5 wt %). The concentration of Ni was rather constant across the coating, whereas the concentration of Cr decreased with increasing distance from the substrate to the outermost part of the coating. Nevertheless, the concentration of Cr was still higher than that for Ni in the outermost part of the coating, suggesting a different diffusion coefficient of Ni and Cr through the coating, respectively. Based on this result, it can be concluded that the diffusion of Ni is much slower in comparison with Cr, since both elements (Ni, Cr) showed the same concentration in the metal matrix (25 wt %), but the Ni and Cr concentration in the outermost part of the coating indicated a higher concentration for Cr (9 wt %) than for Ni (5 wt %), as suggested earlier. Finally, the Super 304H steel with a diffusion coating is shown in Figure 3E. Comparable to the other coatings deposited in this work, the coating showed a sharp interface with the Super 304H substrate. Neither cracks nor delamination nor detachment of the coating were found as well. The thickness of the coatings varied from 50 to 70 µm. The coating in the top part was rich in Al (70 wt %), while the other elements showed a much lower concentration: Fe (15 wt %), Cr (9 wt %), and finally Ni (4 wt %). The middle part of the coating showed a lowered Al concentration (55 wt %) and a higher concentration of Fe (29 wt %), suggesting the formation of the Fe_3_Al–FeAl_2_ phase, while the Cr and Ni concentrations remained the same: 9 wt % and 4 wt %, respectively. In comparison to SANICRO25 steel, the Super 404H material with lower Cr (18.8 wt %) and Ni (8.8 wt %) content in the metal matrix transferred a similar concentration of Cr and Ni to the developed coating as the material with the higher concentration of Cr and Ni (25 wt %). Furthermore, in contrast to SANICRO 25 steel, no Cr rich band was found between the inner part of the coating and the Super 304H substrate.

Below, Figure 4A–E shows enlarged views of the Figure 3A–E in area of the coating–substrate interface with additional EDS results acquired from the area of interest (coating, interface substrate).

### 3.2. Mass Gain Data

Figure 5, Figure 6, Figure 7, Figure 8 and Figure 9 show the kinetic data of the steels covered with the diffusion coatings with RE addition exposed at 650 °C for 1000 h.

After the performance of air oxidation tests, the kinetic results clearly indicate that the coating rich in Al for the steels 16M, T91, and VM12 showed a positive effect. In case of 16M steel (Figure 5), the reference material showed a mass gain of about 70 mg/cm^2^ after 1000 h of exposure at 650 °C. The application of a coating with no RE addition lead to a significant reduction of the mass gain. The addition of RE to the diffusion coating showed not much effect in 16M steel, whereby the coating with the addition of La showed a slightly higher mass gain than the coating with CeO_2_, Gd, and with no addition of RE. No spallation was observed from the base material during the test neither from the coating without RE addition nor from the coating with RE addition. In Figure 6, the kinetic data of T91 steel with 9 wt % Cr with a deposited diffusion coating without and with the addition of RE is shown. In contrast to 16M steel, the coating with the addition of La showed a higher mass gain than the reference sample. In the case of 16M steel, the coating with La showed a higher mass gain than the other coatings with RE addition. The other RE (Gd, CeO_2_) showed a low mass gain (0.2 mg/cm^2^). The coating with no RE addition lost mass due to flaking of the scale. This effect is attributed to brittleness of the Al-rich coatings as suggested by other studies published in the past [20,28]. Furthermore, the addition of RE improves adhesion to the metallic material, since no spallation was observed in the coating 4B (CeO_2_). The coating with the addition of Gd indicated metal loss in the first 100 h. Similar observations regarding a positive effect of the diffusion coatings with the addition of RE was observed as well in VM12 steel, as Figure 7 reports. The highest mass gain was reported for the reference sample, while the lowest was reported for the sample coated by diffusion coating with the addition of Gd. The addition of La and CeO_2_ in case of VM12 steel developed a higher mass gain than the coating with no RE addition. In contrast to the kinetic data presented in Figure 5, Figure 6 and Figure 7, the data presented in Figure 8 and Figure 9 clearly indicate that with the increased Cr content in a steel with higher than 16 wt % of Cr, the effect of a coating rich in Al and with the addition of RE is greatly lowered. Nevertheless, the mass gain obtained by the coated specimens is rather low compared to the reference samples. In case of SANICRO 25 steel with 25 wt % of Cr, the lowest mass gain was observed for the coating with CeO_2_ addition. The other additives showed a higher mass gain than the reference sample, and the highest value was indicated for the coating with the addition of Gd (0.37 mg/cm^2^). Finally, the Super 304H steel showed a slightly better oxidation resistance in a similar way to that of SANICRO25 steel compared with the samples coated with diffusion coatings embedded with the RE.

### 3.3. XRD Analyses

Figure 10, Figure 11, Figure 12, Figure 13 and Figure 14 show XRD analyses of the exposed in air diffusion coatings developed on steels exposed at 650 °C for 1000 h. Each of the figures represents a summary of the XRD patterns for each sample coated by diffusion coating without and with additions of RE.

General observations indicate that all steels covered by pack cementation coatings without and with RE addition developed in air atmosphere predominantly three main phases: Fe_2_O_3_, FeO, and Al_2_O_3_. Furthermore, the peaks originating from the coating were detected as well; for the steels with up to 20 wt % of Cr, the XRD spectra contained the Fe_3_Al, FeAl_3_, and FeAl phases, whereas for the steel with the higher Cr content (SANICRO25), the XRD spectra showed the formation of the Cr-Al, Ce-Fe-Al, Ni_2_Al, and Ni_2_Al_3_ phases. The oxidation process in SANICRO25, besides the formation of Fe_2_O_3_, FeO, and Al_2_O_3_, showed the formation of the W_3_O phase. No phases containing Ni-O were observed during XRD investigations. This leads to the suggestion that the composition of the developed coating based on Fe-Al phases contained a low concentration (low activity) of Ni in the coating matrix, or Ni was localized in the bottom part of the deposited coating when diffused outwardly from the substrate. The observations on the oxidized coatings also showed no influence of RE on the phase constituents within the coatings. However, the addition of RE showed some influence on the crystal parameters of the elemental cell that was observed in Super 304H steel. The observed shifts for the coatings with RE addition compared with the coatings with no RE addition in XRD spectra are presented in Figure 14. The observed peak shifts are correlated with changes in a phase stoichiometry (in this case, the decreased Al concentration in a cell) by doping that influences the lattice parameters according to Vegard’s law [29]. This effect is also recognized mainly due to the difference in ionic radii between the main elements Fe^3+^ (64.5 pm) and Al^3+^ (53.5 pm) and the dopant ions Ce^4+^ (87 pm), La^3+^ (103.2 pm), and Gd^3+^ (93.8 pm), respectively. For sure, the addition of RE dopant to a pack cementation coating rich in Fe and Al provokes a peak displacement, as shown. Based on this result, it can be concluded that the observed shift is higher for the element on a higher oxidation state in this case for Ce, whereas La and Gd dopant showed a similar effect, since the oxidation state for these two elements are the same. Furthermore, it can be concluded that a higher effect in peak displacement is related to oxidation state of a dopant than in ionic radii of the dopant. Nevertheless, the situation presented above was observed in the most clear manner only in the case of Super 304H steel. For the other steels coated with pack cementation coatings doped with RE, no peaks displacement was observed in that scale.

### 3.4. Macro Observations

The exposed samples were captured via a macro lens in order to assess the oxidation process at the macro scale. The results of those observations are shown in Figure 15 for each steel for the samples in the uncoated and coated state without RE and with RE addition, respectively.

The macro observations revealed that the exposed samples in oxidation conditions in general showed good oxidation resistance apart from the uncoated 16M steel, where the oxide scale detached from the metallic substrate (Figure 15). Detachment is a result of a thick oxide scale formation and the generation of strain within the oxide structure and between the metallic core and the oxide scale. This is due to different coefficients of thermal expansion (CTE) of two different materials (here: steel and ceramic). The coated 16M showed a much better oxidation behaviour in the coated state, and there was no detachment of the coating neither with RE addition nor without RE addition. The coated 16M steel changed only the colour of the coatings due to oxidation process of the phase constituents within the exposed coating. The substrates with the higher content than 9 wt % Cr showed no scale or deposited coating detachment. In terms of the microstructure and colour of the surface of the deposited coating, a more uniform colour was observed in the substrate with 18 wt % Cr than in the sample with 25 wt % Cr.

### 3.5. SEM X-ray Mapping Analyses

In this work, the SEM X-ray mapping analyses were conducted only from the cross-sectioned samples of SANICRO25 steel (Figure 16, Figure 17, Figure 18 and Figure 19), since the material underwent the lowest oxidation process according to the kinetic data presented earlier in this work. The materials with Cr concentration up to 12 wt % (16M, T91, VM12) were not analysed extensively in this work (using SEM, EDS X-ray mappings), as similar papers in the past were published [30,31,32]; in addition, Super 304H steel was not analysed in this work either, since the material showed similar behaviour as SANICRO25 steel. Figure 16, Figure 17, Figure 18 and Figure 19 show the SEM micrographs and EDS EDS X-ray mappings on the coated SANICRO25 steel with no RE addition (Figure 16) and the coated SANICRO25 steel with addition of RE elements (Gd (Figure 17), La (Figure 18), CeO_2_ (Figure 19)). The uncoated SANICRO25 steel was not analysed extensively. The readers are encouraged to read the paper presented by Sroka et al. [33], where oxidation studies were carried out for 1000, 5000, and 10,000 h, respectively, at high temperature on SANICRO25 steel. In general, SANICRO25 steel in rich oxygen atmosphere develops a FeCr_2_O_4_ spinel oxide as an outermost layer. Within the material under a longer exposure time, the precipitates rich in Nb ((NbC, NbCrN)–Z phase) and W are present and developed on grain boundaries. These phases are the main elements influencing the solid solution strengthening process in austenitic steels. However, the development of phases on grain boundaries hardly influences the austenitic steel ductility [34]. The performed analyses on the coated sample with diffusion coatings without and with the addition of RE (Gd, La, CeO_2_) indicate some interesting features. The addition of Gd to a pack cementation coating rich in Al showed a slight effect on the microstructure of the coating as compared to the coating with no RE addition (Figure 17 and Figure 18). In both coatings, similar thicknesses were found as well as similar chemical compositions of the exposed coating. However, the Al concentration in the Gd-doped coating is slightly enriched in the outermost part of the coating. There is no enriched layer of Ni in the bottom part of the coating in comparison to the exposed coating with no RE addition. In the Gd-doped coating, a single enriched Cr layer has formed, whereas in the exposed coating with no Gd addition, two Cr enriched bands are observable. This finding is in good agreement with the kinetic data (Figure 8), where the coating with no Gd addition showed a slightly lowered mass gain than the coating with Gd addition. The other elements (Mn, Fe, and O) showed no differences in concentration or the number of layers developed between doped and undoped exposed coating. Furthermore, both coatings showed a large void formation within the structure. The voids probably are the results of the Kirkendall effect, which is a well-known phenomenon resulting from the difference of intrinsic diffusivities of chemical constituents and internal stresses within the coating [35]. The average size of a single void is around 15 μm, while some of the voids have a width of 2–3 μm. These voids are only present in the upper and middle part of the coating, where the Al concentration was the highest. Similar voids were found during oxidation of the ODS alloy MA 6000 alloy [36]; the voids formed by vacancy injection resulting from outward diffusion of mainly Cr seems the most consistent according to our study. In this case, the voids are developed due to the outward diffusion of Al to form an Al_2_O_3_ oxide scale. The number of voids is correlated with the thickness of the oxide scale: the thicker the oxide scale, the higher number of voids formed due to the higher activity of Al in the coating within the outermost region of the coating. The bottom part of the exposed samples (without and with RE addition), enriched with Ni, Cr, and Mn showed no voids formation due to the diffusion of Ni and other metal matrix elements (Cr, Mn) to the coating matrix. Finally, the addition of Gd to an Al-rich diffusion coating deposited on a high Cr steel showed rather no effect on voids formation.

The presented X-ray mapping of the SANICRO25 steel coated by pack cementation coatings with the addition of La and addition of CeO_2_ are shown in Figure 18 and Figure 19 respectively. Both RE additions (La, CeO_2_) strongly influence the coating thickness. This is indicated by a much thinner coating formation compared to that observed when Gd was added to the Al–rich diffusion coating or when the coating was fabricated with no RE addition. This result indicates that the addition of La and CeO_2_ to the Al-rich diffusion coating on high Cr steel (SANICRO25) slows down the diffusion processes strongly, leading to the development of a much thinner pack cementation coating than the coating with no RE addition or with Gd addition. The microstructure of the exposed coatings with the addition of La and CeO_2_ showed similarities: both coatings developed tiny voids in the outermost part, and the number of the formed voids was lower than that observed in Gd doped Al-rich coating, but the reason of the voids formation is the same as in the coating with no RE addition and doped with Gd. However, in this case, the lower number of voids within the coating is related to the lower diffusion of Al from the coating to develop the oxide scale. This further shows that the addition of La and CeO_2_ lowered Al diffusion and slowed down the oxidation process. Thus, in the coatings with La and CeO_2_, there is no observable diffusion of Al from the coating as observed in the coating without the addition of RE or with the addition of the Gd element, which is probably due to the formation of an enriched Cr–Mn layer in the substrate and an Al-rich coating interface or due to the process related with the doped effect with RE elements. On the other hand, a similar enriched Cr–Mn band was observed in the Gd-doped coating–substrate interface along with Al inward diffusion from the deposited coating.

## 4. Conclusions

The aim of this work was to investigate the effect of the addition of Gd, La, and CeO_2_ (up to 1 wt %) to the rich Al diffusion coating developed by the pack cementation technique. Based on the results, the following conclusions are written below:A high degree of protection was found when the coating was deposited on 16M, T91, and VM12 steel.There was not much difference in terms of protection when the coating was doped with RE addition, but pack coatings secure the scale spallation in 16M steel.Not much effect was observed in high Cr steels SANICRO25 and Super 304H with pack coatings with and without the addition of Gd, La, and CeO_2_.EDS X-ray mapping of the coated SANICRO25 steel showed that Gd has no effect on pack cementation coating. However, La and CeO_2_ addition indicated a thinning of the coating.Voids were observed in the outermost region of the coating due to the Kirkendall effect and internal stress formation.The number of voids was much lower than in the coating with no addition and with the addition of Gd.

## Figures and Tables

**Figure 1 materials-14-06801-f001:**
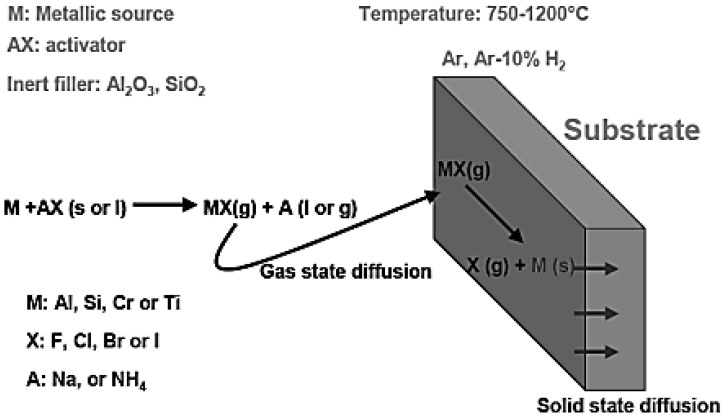
Pack cementation process overview.

**Figure 2 materials-14-06801-f002:**
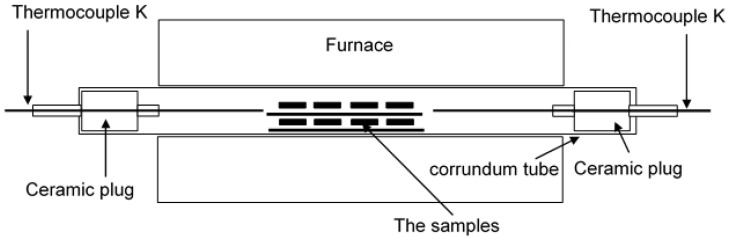
Oxidation test rig used for the evaluation of oxidation resistance at 650 °C for 1000 h.

**Figure 3 materials-14-06801-f003:**
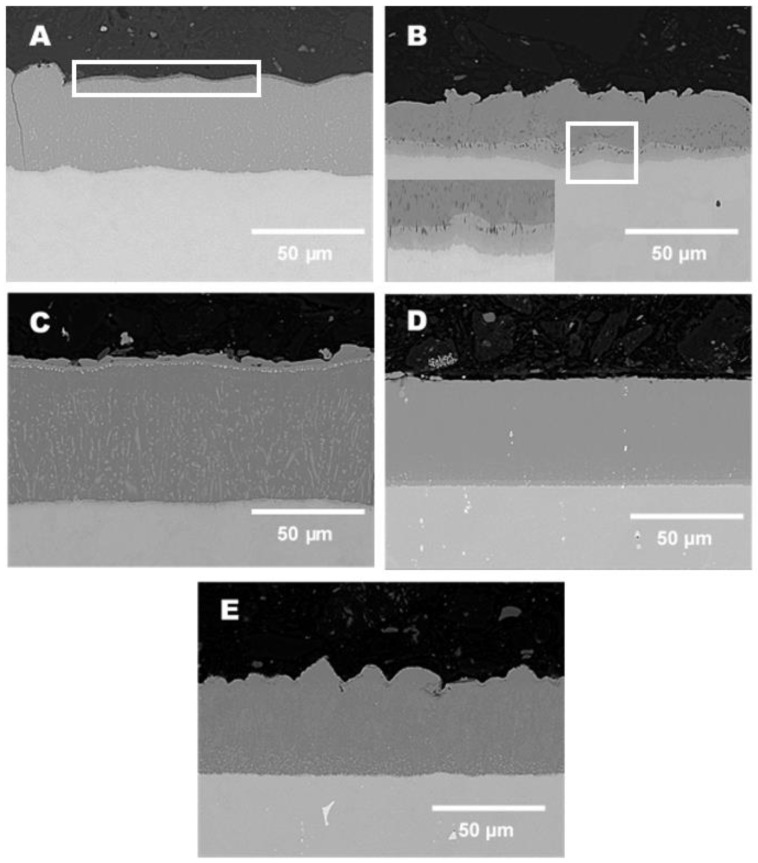
SEM images in BSE mode of the as-deposited diffusion coatings with no addition of RE: (**A**) 16M, (**B**) T91, (**C**) VM12, (**D**) SANICRO25, and (**E**) Super 304H.

**Figure 4 materials-14-06801-f004:**
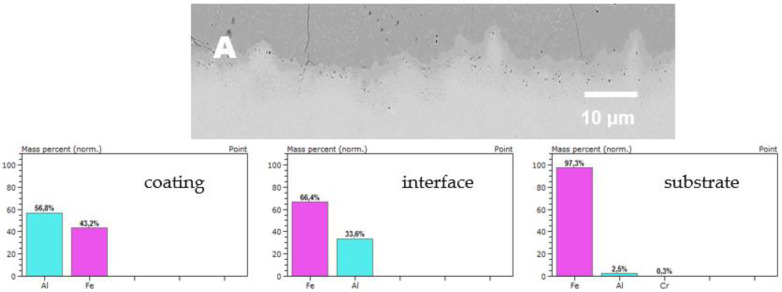
SEM images enlarged area of coating–substrate interface in BSE mode of the as-deposited diffusion coatings with no addition of RE: (**A**) 16M, (**B**) T91, (**C**) VM12, (**D**) SANICRO25, (**E**) Super 304H.

**Figure 5 materials-14-06801-f005:**
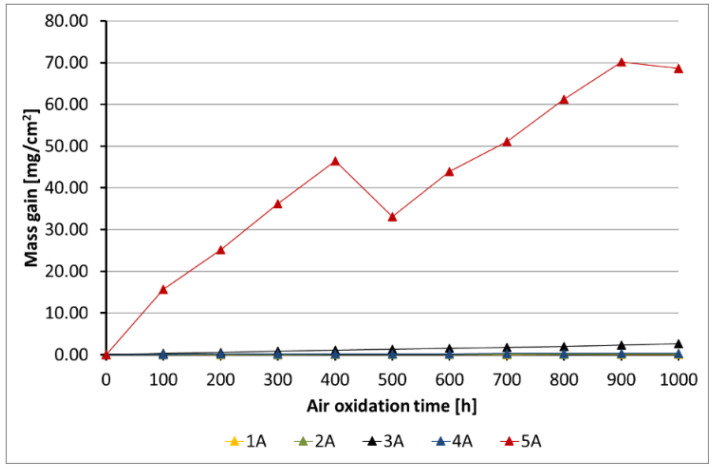
Kinetic data for the 16M steel with diffusion coatings (A): 1 sample with the coating with no RE addition, 2 diffusion coating + Gd, 3 diffusion coating + La, 4 diffusion coating + CeO_2_, 5 reference sample.

**Figure 6 materials-14-06801-f006:**
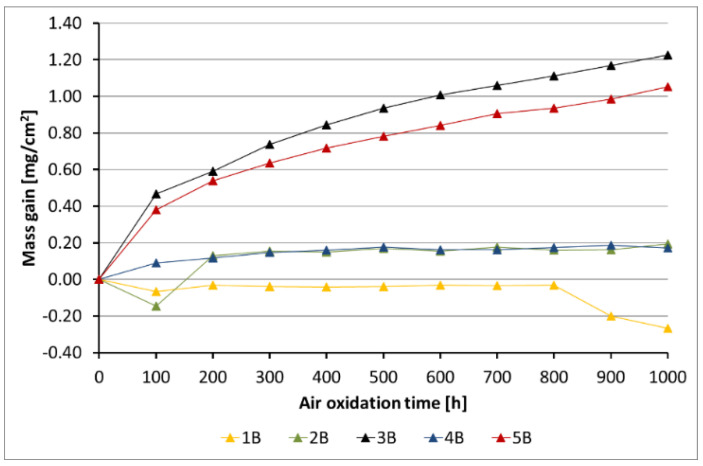
Kinetic data for the T91 steel with diffusion coatings (B): 1 sample with the coating with no RE addition, 2 diffusion coating + Gd, 3 diffusion coating + La, 4 diffusion coating + CeO_2_, 5 reference sample.

**Figure 7 materials-14-06801-f007:**
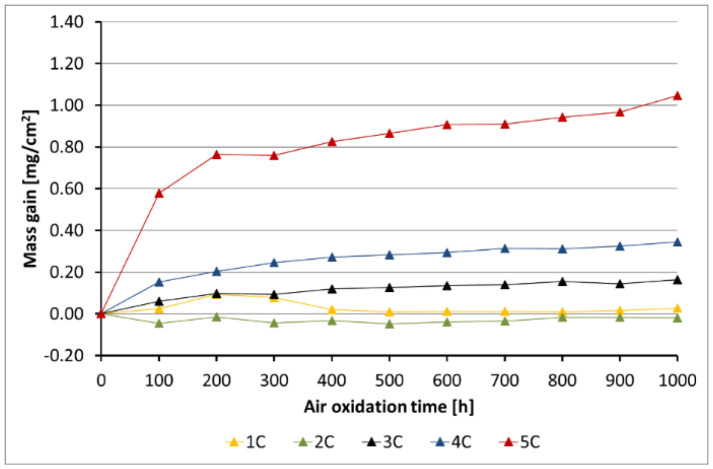
Kinetic data for the VM12 steel with diffusion coatings (C): 1 sample with the coating with no RE addition, 2 diffusion coating + Gd, 3 diffusion coating + La, 4 diffusion coating + CeO_2_, 5 reference sample.

**Figure 8 materials-14-06801-f008:**
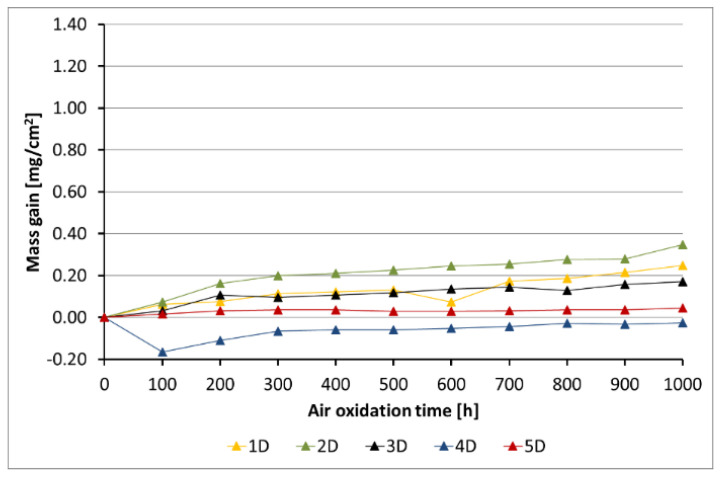
Kinetic data for the SANICRO25 steel with diffusion coatings (D): 1 sample with the coating with no RE addition, 2 diffusion coating + Gd, 3 diffusion coating + La, 4 diffusion coating + CeO_2_, 5 reference sample.

**Figure 9 materials-14-06801-f009:**
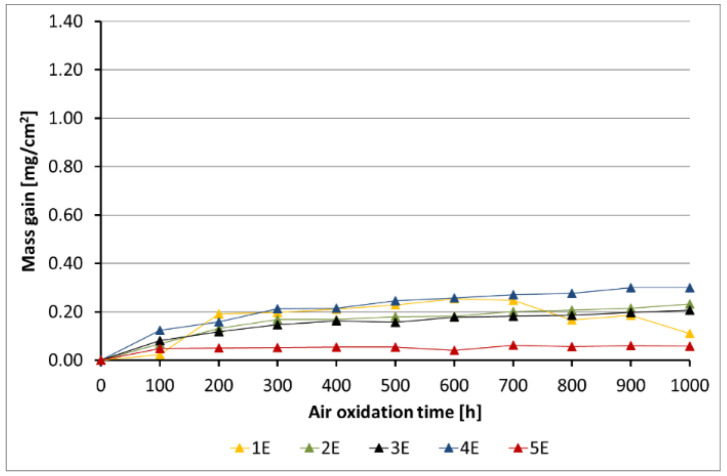
Kinetic data for Super 304H steel with diffusion coatings (E): 1 sample with the coating with no RE addition, 2 diffusion coating + Gd, 3 diffusion coating + La, 4 diffusion coating + CeO_2_, 5 reference sample.

**Figure 10 materials-14-06801-f010:**
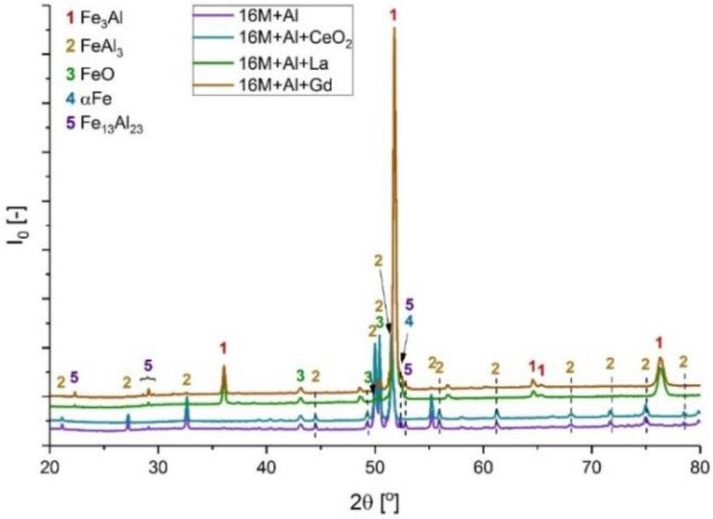
XRD patterns for 16M steel coated with diffusion coating without and with RE exposed at 650 °C for 1000 h.

**Figure 11 materials-14-06801-f011:**
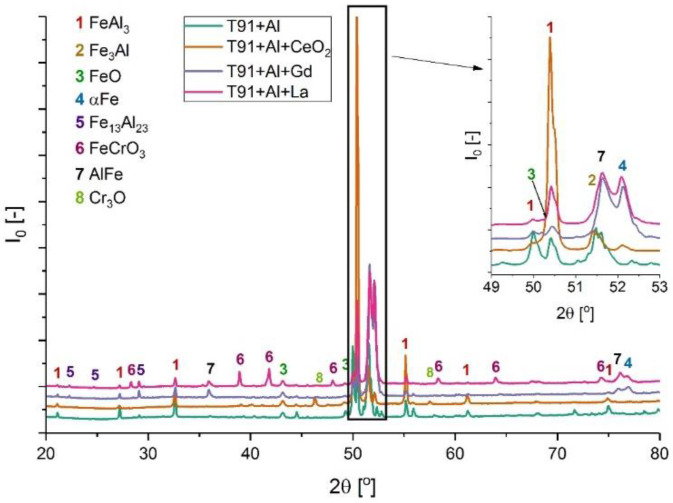
XRD patterns for T91 steel coated with diffusion coating without and with RE exposed at 650 °C for 1000 h.

**Figure 12 materials-14-06801-f012:**
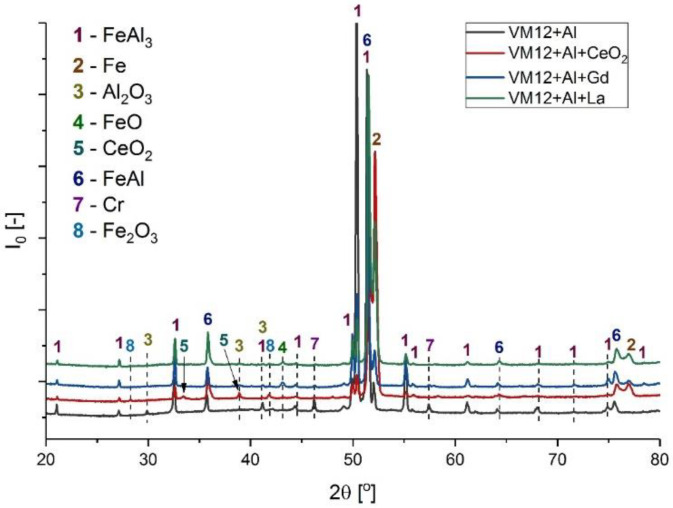
XRD patterns for VM12 steel coated with diffusion coating without and with RE exposed at 650 °C for 1000 h.

**Figure 13 materials-14-06801-f013:**
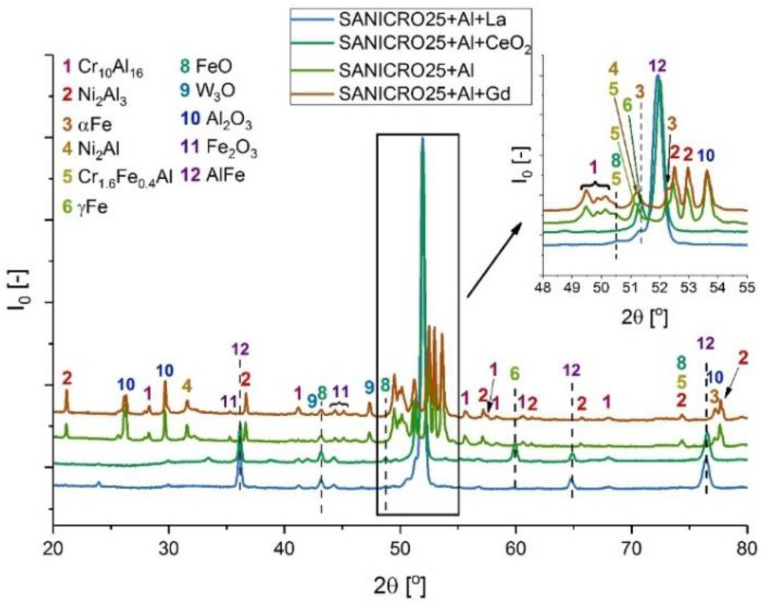
XRD patterns for SANICRO25 steel coated with diffusion coating without and with RE exposed at 650 °C for 1000 h.

**Figure 14 materials-14-06801-f014:**
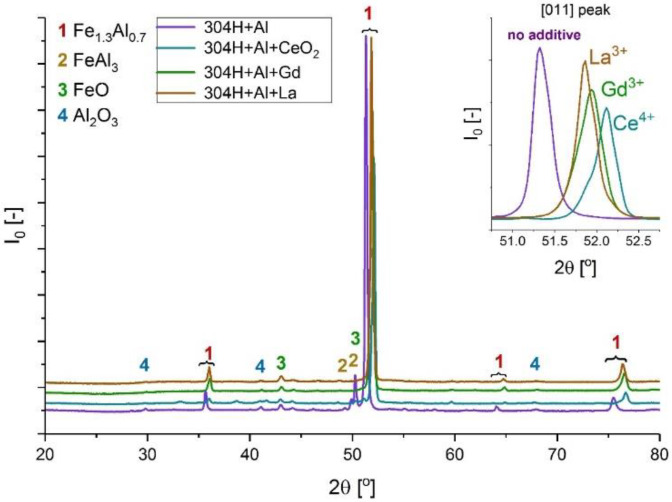
XRD patterns for SUPER 304H steel coated with diffusion coating without and with RE exposed at 650 °C for 1000 h.

**Figure 15 materials-14-06801-f015:**
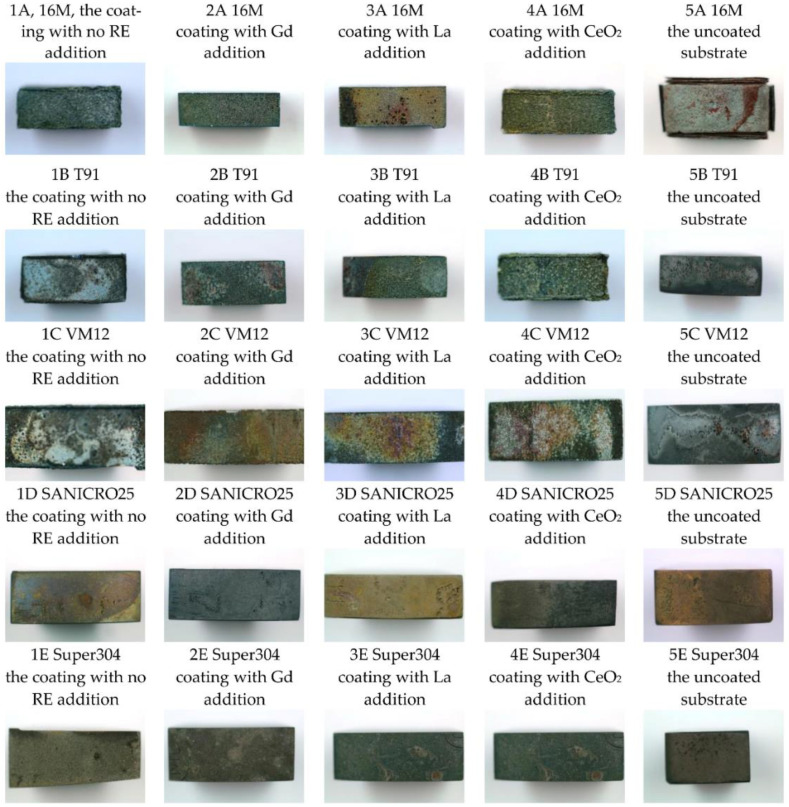
Macro images of the steels oxidised at 650 °C for 1000 h (the uncoated and the coated materials).

**Figure 16 materials-14-06801-f016:**
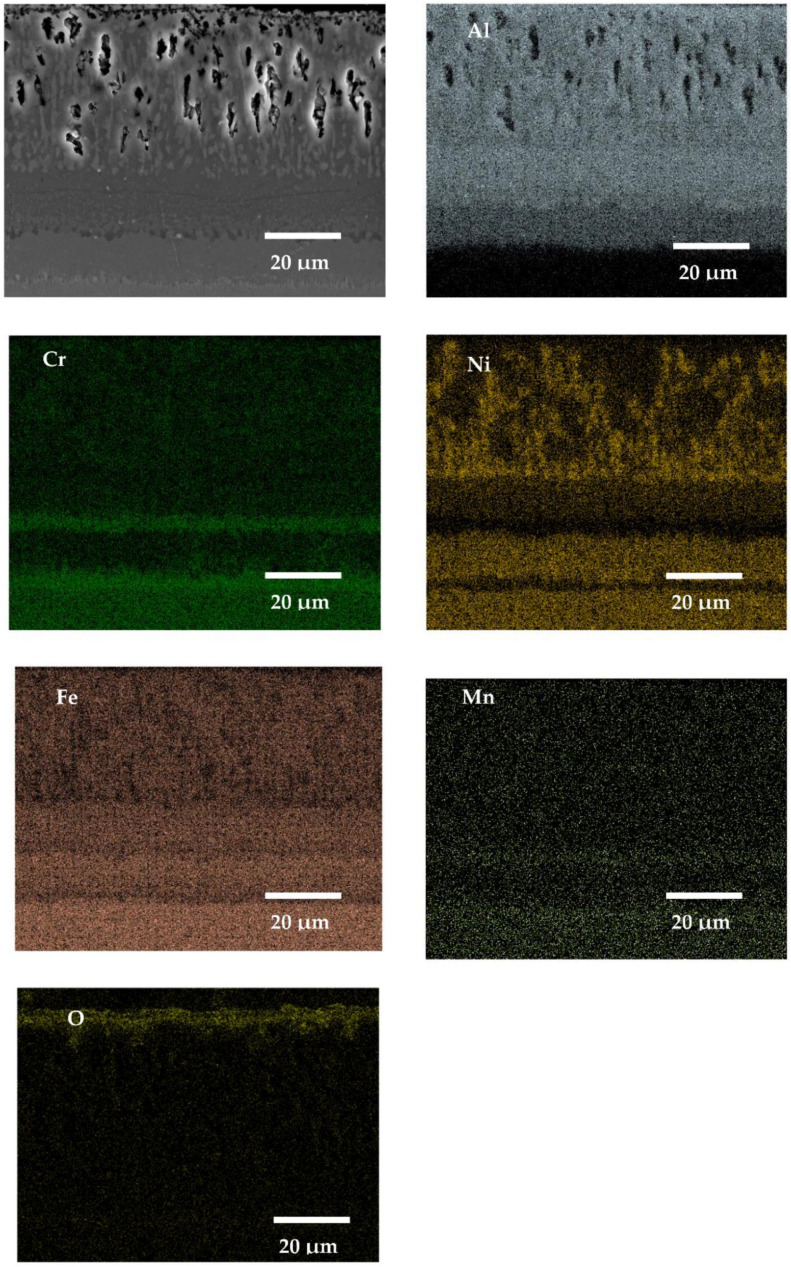
SEM image and X-ray mapping for SANICRO25 steel with an Al pack coating with no RE addition exposed for 1000 h at 650 °C in air atmosphere.

**Figure 17 materials-14-06801-f017:**
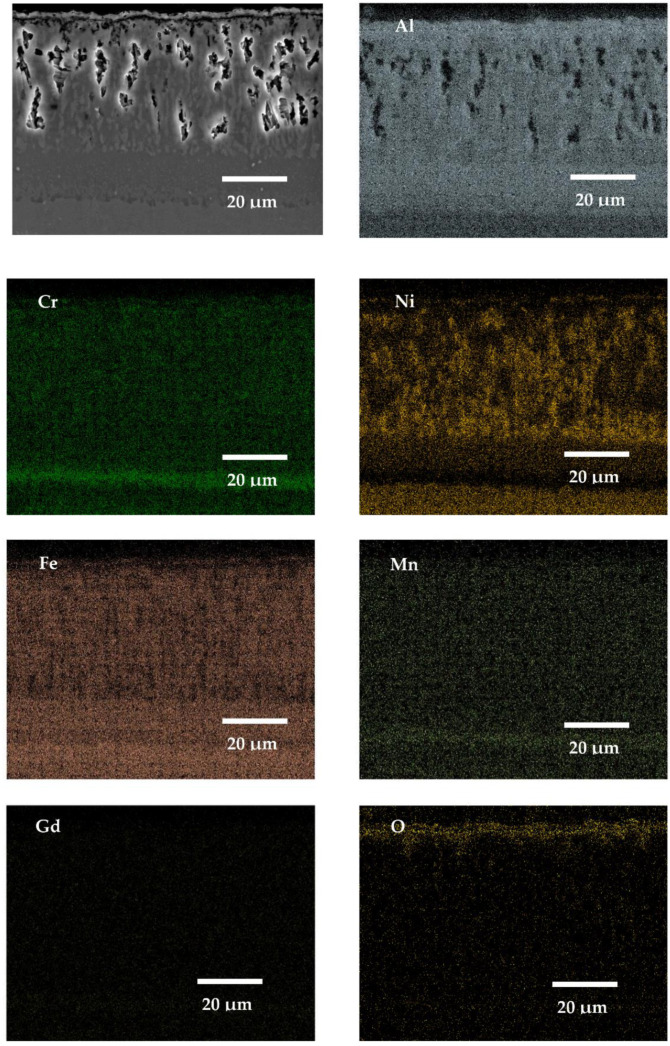
SEM image and X-ray mapping for SANICRO25 steel with Al pack coating with Gd addition exposed for 1000 h at 650 °C in air atmosphere.

**Figure 18 materials-14-06801-f018:**
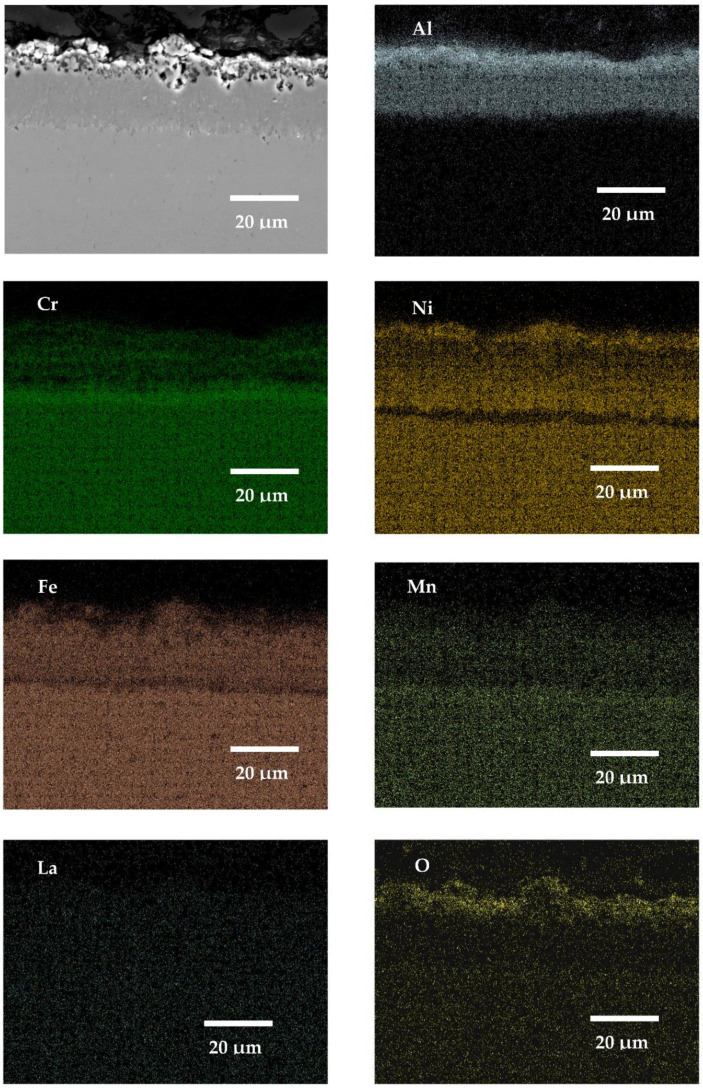
SEM image and X-ray mapping for SANICRO25 steel with Al pack coating with La addition exposed for 1000 h at 650 °C in air atmosphere.

**Figure 19 materials-14-06801-f019:**
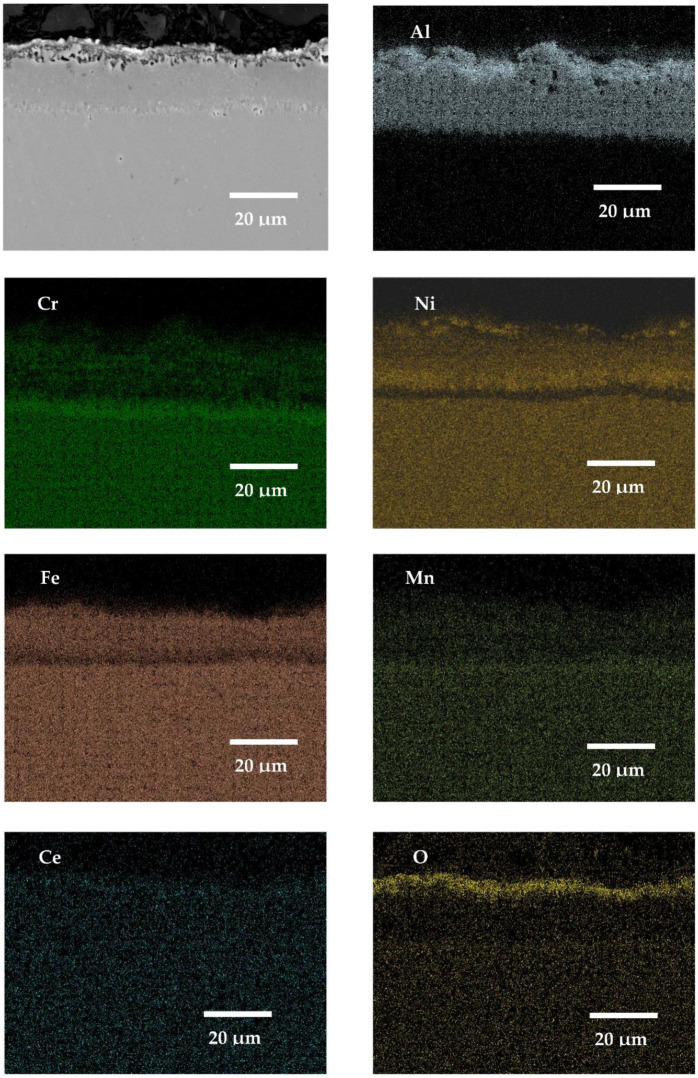
SEM image and X-ray mapping for SANICRO25 steel with Al pack coating with CeO_2_ addition exposed for 1000 h at 650 °C in air atmosphere.

**Table 1 materials-14-06801-t001:** Chemical composition of the substrates used in this work in wt %.

Steel	C	Si	Mn	P	S	Ni	Cr	Nb	Others:
16M	0.12	<0.35	0.4	0.025	0.025	-	0.3	-	Mo: 0.3
T91	0.12	0.3	0.5	0.02	<0.01	-	9.0	-	Mo: 1
VM12	0.12	0.5	0.5	-	-	0.4	11.0	0.06	W: 1.5 B: 0.001 N: 0.02 V: 0.2 Mo: 0.4
Super 304H	0.07	0.23	0.8	0.010	0.009	8.7	18.8	0.27	Cu: 2.68
SANICRO25	<0.1	0.2	0.5	<0.025	<0.025	25.0	25.0	0.5	Co: 1.5 W: 3.6 N: 0.23 Cu: 3.0

**Table 2 materials-14-06801-t002:** Sample codes used in this work.

No.	Code	Steel Grade	Coating Symbol
1	A	16M	1 (Coating with no RE addition)
2			2 (Coating with Gd)
3			3 (Coating with La)
4			4 (Coating with CeO_2_)
5			5 (Reference sample)
6	B	T91	1 (Coating with no RE addition)
7			2 (Coating with Gd)
8			3 (Coating with La)
9			4 (Coating with CeO_2_)
10			5 (Reference sample)
11	C	VM12	1 (Coating with no RE addition)
12			2 (Coating with Gd)
13			3 (Coating with La)
14			4 (Coating with CeO_2_)
15			5 (Reference sample)
16	D	SANICRO 25	1 (Coating with no RE addition)
17			2 (Coating with Gd)
18			3 (Coating with La)
19			4 (Coating with CeO_2_)
20			5 (Reference sample)
21	E	Super 304H	1 (Coating with no RE addition)
22			2 (Coating with Gd)
23			3 (Coating with La)
24			4 Coating with CeO_2_)
25			5 (Reference sample)

## Data Availability

The study did not report any data in publicity.

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
