# Peer review of "Preliminary Studies on Rare Elements Addition and Effect on Oxidation Behaviour of Pack Cementation Coatings Deposited on Variety of Steels at High Temperature"

_materials, 2021, doi:10.3390/ma14226801_

Round 1

Reviewer 1 Report

The paper entitled “Preliminary studies on rare elements addition and effect on oxidation behavior of pack cementation coatings deposited on variety of steels at high temperature” presents the effect of rare elements addition and pack cementation coatings deposited on the oxidation behavior of various steels. Specifically, the research object has five kinds of steels, namely 16M, T91, VM12, 304H and SANICRO25. The five variable factors are three rare elements (Gd, CeO2, La) added under the coating, only coating and no coating. The microscopic morphology, XRD pattern and kinetic data of the above five steels under five variable factors were studied to analyze their oxidation behavior at 650 ℃ for 1000 h. The research content of the oxidation behavior of various steels under various influencing factors is in the scope of Materials and interesting. However, after reading the paper I have following comments/suggestions, which require the author to make major revisions seriously in the manuscript.

  1. The main conclusions should be stated concisely in the abstract results, and the content of the test should not be described too much.
  2. In the first paragraph of the introduction, 304H and 316L steels are mainly described, but later, the research is conducted on the five steels of 16M, T91, VM12, 304H and SANICRO25. It is necessary to explain the reasons for the selection.
  3. In the introduction, the research significance of this paper is lacking, and supplementary explanation is needed.
  4. In the first paragraph of the materials, the Cr element was introduced emphatically, and it was necessary to explain why so much attention was paid to the content of Cr in the five steels.
  5. In line 117 of the paper, the average particle size of the ground powder particles is 50 µm. It is recommended to add a particle size distribution diagram of the powder for illustration.
  6. In line 130-133 of the paper, heat treatment for 6-30 hours will affect the microstructure of the matrix, cause grain coarsening and carbide deposition, and affect its mechanical properties. Why can heat treatment for 12 hours avoid the above disadvantages?
  7. It is recommended that all tables in the paper adopt a three-line layout.
  8. In Figure 2, the hearth in the picture should be a corundum tube, but it is marked as "Al tube".
  9. In line 212 of the paper, the figure number should be Figure 3, but it was mistakenly written as "Figure 1".
  10. In “3.1 As-deposited diffusion coatings”, there are a lot of composition analysis on various steels, and it has been mentioned many times based on EDS analysis, but there are only micro-topography images at the corresponding positions, and there is no corresponding EDS images. The author needs to carefully modify this section and add the corresponding EDS images for explanation.
  11. In line 218 of the paper, it is mentioned that the Cr layer is 2 um thick and the content is 20 wt.%. It is recommended to add a partial enlarged view and an EDS analysis diagram in Figure 3 to prove the above results.
  12. In line 242 of the paper, there is an error in the written steel grade, which should be "Super304H".
  13. In line 298-330 of the paper, there is no in-depth analysis of the reasons for the phenomena in Figures 4 to 8, and the results are simply described according to the trends of the curves, like a technical report.
  14. In line 339 of the paper, figure 10 has only the name of the image, but no image, please add.
  15. In sections 3.2-3.4 of the paper, the images are all above the main text, which leads to unsuccessful reading. It is recommended that the author place the image below the corresponding text when writing a scientific paper.
  16. The layout of Figures 14-18 is too confusing and takes up a lot of space. It is recommended to re-typesetting, and strive to be compact, concise and beautiful.
  17. It can be seen from Figure 14 that the oxidation degree of 1A and 4A is very serious, and the difference is not much compared with that of uncoated 5A. These phenomena need to be explained. At the same time, Figures 15-18 only briefly explained the phenomenon, and did not carefully analyze the content of the pictures. It is necessary to explain the difference instead of simply describing the phenomenon.
  18. In “4. Conclusions”, the results need to be stated separately to make them clearer, mainly conclusive sentences, and should not explain too many reasons.

Author Response

Dear Reviewer

Thank you for your comments in the paper, I hope we clarified your doubts.

Sincerely,

Tomasz

Reviewer 2 Report

Authors Dudziak et al. present the results of studies on the effect of rare-earth elements on the oxidation behavior of pack cementation coatings on various steels at high temperature. The authors clearly detail the current state of research in this area and provide a clear reasoning for the appropriateness of these studies. These studies are quite relevant. Nevertheless, I cannot recommend the manuscript for publication in its present form. Below are my comments and suggestions.

(1) I recommend presenting Fig. 1 in higher quality.

(2) The reference should be added to the caption of Fig. 1.

(3) In my opinion, a brief description of the methods and equipment used to characterize the samples obtained needs to be added to Section 2.

(4) Line 118-119 “The same mixture of powders contained Al (8%), Al2O3 (88%) and a halide salt AlCl3 (4 %) and.”  Is some component of the mixture missing?

(5) I suggest reorganizing Table 2 or providing more detailed comments in the text. At first glance, it appears that, for example, the 16M steel coating contained no rare-earth cation additions, and the T91 coating contained only Gd, etc. And only further  it becomes clear that for each steel sample all the coatings specified in Table 2 were realized.

(6) Section 3.1: how was the thickness of coatings measured.

(7) Lines 181-183 “The EDS analysis performed on the coated 16M steels showed mostly a single phase coating with the outermost thin layer with a thickness of 1– 2 μm, the major layer thickness reached around 60 μm”. Clarify, please, what an “outermost” layer is and what the authors mean by “basic” layer. 

(8) As can be seen from the images in Fig. 3, the coatings are heterogeneous in composition (the coatings show light gray (3A, C, E) and black (3B) inclusions). What is the composition of these inclusions?

(9) In Fig. 3 a clear straight steel-coating boundary is fixed only for the 3D image. In other cases, especially for 3B, it is curved. Briefly explain the reason for this phenomenon.

(10) For the benefit of readers, I recommend that the surface of the steel, the main layer, the outer layer and the intermediate layer, if any, be marked in Figure 3. As well as the composition.

Author Response

(The authors gave the same response as above.)

Round 2

Reviewer 1 Report

This manuscript has met the requirements for publication and is recommended for acceptance.